# A Lifetime of a Dispenser-Release Rates of Olive Fruit Fly-Associated Yeast Volatile Compounds and Their Influence on Olive Fruit Fly (*Bactrocera oleae* Rossi) Attraction

**DOI:** 10.3390/molecules28062431

**Published:** 2023-03-07

**Authors:** Maja Veršić Bratinčević, Ana Bego, Ivana Nižetić Kosović, Maja Jukić Špika, Filipa Burul, Marijana Popović, Tonka Ninčević Runjić, Elda Vitanović

**Affiliations:** 1Department of Applied Science, Institute for Adriatic Crops and Karst Reclamation, Put Duilova 11, 21000 Split, Croatia; 2Ericsson Nikola Tesla, Poljička Cesta 39, 21000 Split, Croatia; 3Center of Excellence for Biodiversity and Molecular Plant Breeding (CoE CroP-BioDiv), Svetošimunska Cesta 25, 10000 Zagreb, Croatia; 4Department of Plant Sciences, Institute for Adriatic Crops and Karst Reclamation, Put Duilova 11, 21000 Split, Croatia

**Keywords:** olive, olive fruit fly, yeast, semiochemicals, volatile release rate, slow-release dispensers, HS-GC/FID

## Abstract

The objective of this study was to evaluate the release rate, duration, and biological efficiency of yeast volatile compounds associated with olive fruit flies in slow-release dispensers, polypropylene vials, and rubber septa attached to yellow sticky traps under different environmental conditions in order to protect the environment, humans, and nontarget organisms. Isoamyl alcohol, 2-octanone, and 2-phenethyl acetate were placed in dispensers and tested over a four-week experiment. The weight loss of the volatile compounds in both dispensers was measured, and a rapid, inexpensive, and simple HS-GC/FID method was developed to determine the residual amount of volatiles in the septa. 2-Phenethyl acetate stood out in the rubber septa and showed a statistically significant difference in the release ratio compared to the other volatiles under all conditions tested. Our results showed that the attraction of olive fruit flies increased with decreasing concentrations of the tested volatiles. Regarding the number of flies attracted by rubber septa containing 2-phenethyl acetate, significantly better results were obtained than for septa containing isoamyl alcohol and 2-octanone, in contrast to the attraction of olive fruit flies to polypropylene vials containing these compounds but without significant difference. Since the presence of all tested chemicals was detected during the experiment, this opens the possibility of using more environmentally friendly and cost-effective dispensers with a significantly lower amount of semiochemicals.

## 1. Introduction

Introducing new strategies or improving existing ones beyond insecticides is critical to maintaining agricultural production and improving our environment and health. Biotechnical methods play a particularly key role in the control of Tephritidae and are used to a much greater extent in the control of Tephritidae than in any other order or family of insect pests [1]. Among the biotechnical methods, the use of semiochemicals is playing an increasingly important role [2,3,4].

Semiochemicals are widely used to monitor and detect insect pests and to suppress populations by attracting and killing, mass trapping, mating disruption, and push–pull strategies [2,5,6]. These are substances or mixtures that allow for the manipulation of insect behavior, both from the same species and from different species, as the release of a single compound from an organism can elicit a behavioral or physiological response [4]. Therefore, behavioral manipulation, as a management tool, could be a great alternative for Tephritidae control, as there is a trend towards reducing the use of insecticides (European Green Deal) and increased resistance towards insecticides, which results in a great need for finding novel, ecologically based measures. This is a promising area, especially in the case of the olive fruit fly (OFF) *Bactrocera oleae* Rossi (Diptera: Tephritidae), the most economically important pest of olive crops worldwide, which can cause yield losses in both table olive and olive oil production [7].

Semiochemicals represent a wide range of volatile and nonvolatile organic substances or mixtures that depend on various physicochemical properties, such as the nature of the semiochemical itself, its volatility and solubility, and the lifetime of the semiochemical in the environment, highlighting the influence of temperature, which can allow increased diffusion of semiochemicals, as one of the most important abiotic factors [6]. In the context of monitoring and control of OFF, several studies have shown that OFF is indeed attracted to selected semiochemicals such, as plant volatiles [8,9,10,11,12,13,14] and yeast volatiles [15,16,17], the emission of which serves as a volatile signal for insect foraging [18]. Vitanović et al. [16] reported the attraction of various yeast species isolated from different habitat sources, such as infested olive fruit, the surface of OFFs, the guts of larvae and adult insects, and other insect species to OFF. Upon further examination of these yeast species, Vitanović et al. [15] noted the presence of several volatile compounds, including isoamyl alcohol, 2-phenethyl acetate, isobutanol, isobutyl acetate, 2-phenethyl acetate, and 2-octanone. These volatile compounds (termed olive fruit fly (OFF)-associated yeast volatile compounds) were even more effective than yeast formulations alone, as well as the control treatment, a non-yeast volatile compound.

The behavioral manipulation of insect pests requires the controlled release of semiochemicals that mimic natural processes. For such methods of behavioral manipulation to be effective, it is critical to find the most appropriate medium for release. This can be achieved by selecting a releasing device that allows for the sustained release of optimal concentrations of selected semiochemicals to elicit the desired insect responses over the intended time, achieve a repeated and stable release rate, achieve replication-induced behavior, and protect the semiochemicals used from UV light, oxygen, or reactions with dispensers; thus, the devices should tend to be inexpensive and environmentally friendly [19]. The dispensers for the slow release of semiochemicals are mostly polyethylene and polypropylene bags, tubes, bottles, beads, glass, rubber septa, tubes and rods, membranes, polyvinyl chloride, spiral polymer dispensers, cellulose, polyester, spiral polymer dispensers, etc. [6,19]. The objective of slow-release devices is to protect semiochemicals from degradation by oxygen and/or UV activity. In addition, certain conditions must be met, including the controlled release of the chemicals at the correct concentration for insect detection, efficiency, and repeatability throughout the application period [6].

In view of this, in our previous study, the presence of OFF-associated yeast volatile compounds was tested over seven days in an olive canopy and analyzed by HS-SPME-GC/FID [17]. The results showed that all of these compounds were present in the dispensers during the experiment in the olive grove. However, to our knowledge, the release rate of OFF-associated yeast volatile compounds has not been studied, and there is little information on their behavior, both under controlled conditions and in the field. To help maximize the effect of the potential attractant for olive fruit fly, our study presents laboratory and field research to better understand and apply semiochemicals, particularly OFF-associated yeast compounds.

Thus, the aim of this research was to investigate the release rate of OFF-associated yeast volatile compounds in two types of slow-release dispensers and relate them to their biological efficiency in attracting OFF under different environmental conditions. In addition, the duration of the slow-release dispensers containing volatiles was tested for possible longer-term use in olive groves.

## 2. Results and Discussion

The four-week experiment, conducted in October 2021, testing three yeast volatile compounds associated with the olive fruit fly (OFF) and one non-yeast volatile compound, was divided into two parts, as shown in Figure 1. The presence of the tested volatiles in the slow-release dispensers, rubber septa (RS), and polypropylene vials (PPVs) under different environmental conditions was determined by mass measurement and headspace gas chromatography with flame ionization detection (HS-GC/FID), while the attraction of the olive fruit fly to the tested volatiles was monitored for both slow-release dispensers attached to yellow sticky traps (YSTs).

### 2.1. Release Rates of the Tested Volatile Compounds

#### 2.1.1. Monitoring the Mass Change in the Volatile Compounds in the Slow-Release Dispensers

The individual monitoring of the volatile residues of three OFF-associated yeast volatile compounds, namely, isoamyl alcohol, 2-octanone, and 2-phenethyl acetate, and one non-yeast volatile compound, n-hexane, (data presented in Table 1), and consequently release kinetics monitoring were performed in October 2021. The presence of volatile compounds was determined by weight loss measurement and gas chromatography.

The release rate of all volatiles was monitored under all conditions, both controlled and field conditions, with both types of dispensers, RS and PPV, by measuring the change in weight starting from the initial weight of the dispenser, the weight of the dispenser with the addition of the volatile, and the weight of the dispenser during the 4 week experiment (Appendix A). This test was performed to determine the weight change over 4 weeks to obtain data on the possible presence of volatiles in the dispenser during the experiment to adjust the time of use of the dispenser in the field for the attraction of OFFs. As can be seen from the results (Appendix A), there were differences between the tested volatiles under the different conditions in the different passive dispensers. Thus, as expected, there was a trend of a decrease in mass under all conditions tested, while the greatest changes occurred in the measurement of weight loss in RS under room conditions, with the results reaching negative values (Appendix A) for isoamyl alcohol, 2-octanone, and n-hexane. It can be concluded that, in addition to the material of which the dispenser is made, the possibility of moisture absorption, expansion, or contraction of the dispenser itself, due to the temperature or pressure, could affect the instability in the measurement of the mass of the septum. Because the initial volume of the volatile compounds in the PPV dispenser was 20 times greater, the measurement of weight loss over the 4 week experiment was easier to perform. The correlation between the mass change in the rubber septa and the mass change in the polypropylene vials is shown in Table 2. It was shown that there was a strong positive correlation when testing 2-phenethyl acetate and n-hexane in septa and vials under all conditions, while 2-octanone showed a positive correlation only under controlled conditions, those of the environmental chamber. In contrast, isoamyl alcohol showed a weak correlation without statistical significance. Isoamyl alcohol and 2-octanone, loaded in RS, showed the greatest decrease in weight loss in the environmental chamber and loaded in PPV in the field. Contrary to expectations, 2-phenethyl acetate loaded in RS and PPV evaporated faster in the field, concluding that the compounds evaporate faster in the field than under controlled conditions due to the influence of abiotic factors. As expected, the release rate of n-hexane was the fastest under all conditions tested and in both dispensers, as it has the lowest molecular weight value of all volatile compounds tested. In addition, the increased volatility of n-hexane is due to the fact that it is a very nonpolar compound without polar functional groups (Table 1). 

Kuenen and Siegel [21] used rubber septa in their experiment to determine the release rates of volatile compounds and synthetic pheromone of the Oriental fruit moth and showed variability in the release rate as a function of solvent type and volume. Butler and McDonough [22] used alcohol and acetate molecules detected as sex pheromones of moth species applied to rubber septa and showed that the molecular size was one of the most crucial factors affecting the release ratio.

Although it was envisaged that the volatiles would exhibit the fastest rate of evaporation under the highest temperature conditions, in our study the slowest rate of decline was observed when the dispenser masses of all volatiles tested were measured in the environmental chamber, where there was no wind and other climatic factors that could affect the faster evaporation of the volatiles. This is likely due to the interaction of the humidity and temperature values, which is why it was difficult for the volatiles to achieve a constant release. Moreover, when the percentage of humidity is high, there may be a change in mass due to the presence of various other factors, such as the accumulation of dust and other impurities and/or contaminants on the dispensers and the influence of other abiotic factors, which makes it difficult to maintain a constant release rate (zero-order kinetics), which is crucial for determining the duration of the dispenser in field experiments and depends on the climatic conditions, the type of dispenser, and the type of molecule tested [6,23]. Accordingly, our results showed that this type of measurement technique is not precise enough, especially for smaller amounts of volatile compounds in passive dispensers, and the main drawback is precisely the insufficient precision and accuracy for determining the release rates of the tested volatiles. From the obtained results (Appendix A), it can be concluded that the presence of volatiles can be determined by mass measurement, especially for dispensers with a larger volume of volatile compounds. However, this does not mean that a smaller amount of the tested volatile compounds filled in smaller dispensers were not present in the dispensers, so one of the more accurate analytical techniques would have to be used for a more accurate determination. In addition, a higher concentration and volume of volatiles do not necessarily attract more OFFs.

#### 2.1.2. HS-GC/FID Analysis of Volatile Compounds

In addition to weighing the dispensers containing the volatiles, an HS-GC/FID analysis was performed to determine the residual amount of each volatile compound (Figure 2 and Figure 3), as the results of the weight change test on RS were not accurate, mainly due to the low weight of the dispenser itself and the added volume of volatile compounds. However, the presence of all tested volatiles in the PPV during the study was not in question, as a 20-fold larger volume was used, in addition to the fact that all of the volatiles were visible; this was also confirmed by the mass measurement (Appendix A), where none reached a negative value, so it was not necessary to detect their presence by gas chromatography. The results are presented as peak area results (Figure 2 and Figure 3) and expressed as a mass concentration calculated from calibration curves (Appendix A). Three OFF-associated yeast volatile compounds, isoamyl alcohol, 2-octanone, and 2-phenethyl acetate, were tested under controlled conditions and in the field (Figure 2 and Figure 3), while n-hexane, a non-yeast volatile compound, was tested only under field conditions to detect its presence during the 4 week experiment (Figure 3). The release rate of the volatile compounds over time followed an exponential function, described by the following formula, where a represents the initial value, b is the factor of decay, and y is the function’s variable:y = abx (1), where 0 < b < 1.(1)

The measured values were fitted to an exponential function using nonlinear least squares [24].

Vitanović et al. [15,16] and Bego and Burul et al. [17] also demonstrated in their previous studies that some OFF-associated yeast volatile compounds attract OFFs. Therefore, it was necessary to test the individual volatile compounds under controlled conditions and in the field, as well as to test the performance of the diverse types of dispensers to improve our knowledge of the chemical properties of the volatile compounds. As far as we know, the release rate of the tested volatile compounds under controlled conditions and in olive groves has not yet been studied. The results of our study showed the presence of all tested compounds over 4 weeks (Figure 3 and Appendix A).

Three OFF-associated yeast volatile compounds were tested under controlled conditions and in the field (Figure 2, Figure 3 and Appendix A), whereas all tested compounds (3 OFF-associated yeast volatile compounds and n-hexane as a non-yeast volatile compound) were analyzed in the field (Figure 3). Under controlled conditions in the environmental chamber, all tested volatile compounds evaporated faster than in the field, as shown in Figure 2. Under all conditions tested, the levels of isoamyl alcohol and 2-octanone decreased sharply during the first week of the experiment, although they were still present. Nevertheless, the presence of all tested volatiles was confirmed and expressed as a percentage of the initial volume of the tested volatile compound under all tested conditions, and they varied from 0.013% in the environmental chamber to 0.862% in the olive grove for isoamyl alcohol, from 0.817% in the environmental chamber to 6.501% under room conditions for 2-octanone, and from 3.133% under room conditions to 22.514% in the environmental chamber for 2-phenethyl acetate, while n-hexane was tested only in the field and reached 3.228% of the initial volume (Appendix A). 2-Phenethyl acetate reached the same level as the other volatile compounds in the environmental chamber after the 2nd week of the experiment, and a slower decline was observed in the field and under room conditions. Overall, 2-phenethyl acetate proved to be the most stable compound with the slowest evaporation under all conditions. In addition, isoamyl alcohol evaporated the fastest under all conditions, followed by 2-octanone (Figure 2 and Figure 3). As expected, the stability of the volatile compounds of the OFF-associated yeast over the 4 week experiment was the most favorable under the controlled laboratory conditions (expressed as the percentage of tested volatile compounds from the initial volume in slow-release dispensers: 0.291% for isoamyl alcohol, 0.978% for 2-octanone, and 4.135% for 2-phenethyl acetate), which is probably due to the less variable climatic parameters, while the volatile compounds in the field were exposed to the influence of abiotic parameters.

To observe the influence of the different climatic parameters on the presence of volatile compounds in the rubber septa over the 4 week experiment, a correlation was made, as shown in Table 3. The concentration of volatile compounds correlated with temperature, humidity, and air pressure under all conditions (i.e., under controlled conditions and in the field experiment), with the addition of precipitation, cloud cover, and wind in the field experiment. Of all the climatic parameters tested, temperature, precipitation, cloud cover, and wind showed a negative correlation with the concentration of volatiles tested, while humidity and air pressure were positively correlated. As shown in Table 3, among all tested OFF-associated yeast volatiles, only 2-phenethyl acetate showed a significant negative correlation with temperature under all environmental conditions, both controlled and field conditions. A comparison of the concentration of OFF-associated yeast volatiles with other climatic parameters showed no statistically significant results, either positive or negative. In addition, there was a statistically significant negative correlation between the temperature and concentration of the n-hexane tested in the field.

With the exception of isoamyl alcohol, which could only be detected in trace amounts under field conditions, all tested compounds were quantified under all environmental conditions using HS-GC/FID, which confirmed their presence throughout the duration of the experiment. For this reason, we concluded that this technique is the better choice, especially when compared to the results obtained by volatile mass change measurements. This analytical technique is much more accurate, sensitive, and precise and allows for the detection of components that are not easy to measure with an analytical balance due to the use of small volumes and other factors that may affect the result (climatic parameters, accuracy and precision of the balance itself, conditions under which the weighing is performed, etc.).

### 2.2. Field Bioassay: Olive Fruit Fly Attraction to OFF-Associated Yeast Volatile Compounds in an Olive Grove

The attraction of *Bactrocera oleae* to three OFF-associated yeast volatile compounds loaded in RS or PPV and attached to YSTs was studied in an olive grove in October 2021 (Figure 4 and Figure 5). The total number of flies caught on YSTs with RS loaded with each of the OFF-associated yeast volatiles tested is shown in Figure 4. The figure shows that YSTs with RS, loaded with 2-phenethyl acetate, were the most attractive to olive fruit flies of all the traps tested. It is well known that ripening and fermenting fruits emit various volatiles, especially esters and alcohols. Fermentation volatiles, such as esters, serve as attractants for many insect species [25], so it was expected that 2-phenethyl acetate would be one of the most attractive volatiles to olive fruit flies in the field. During the first week of the study, the attraction of all the traps tested was similar, and there were no differences in the number of olive fruit flies caught among them. From the second week until the end of the experiment, the attractiveness of the YSTs with RS loaded with 2-phenethyl acetate increased steadily, and the number of flies caught was twice that of the YSTs with RS, loaded with 2-octanone or isoamyl alcohol, and the control traps (Figure 4). During the study, no differences were observed between male and female catches in any of the traps examined (Figure 6).

For each YST pair containing RS with two different OFF-associated yeast volatiles, a Mann–Whitney U test was performed to determine if there were significant differences in attraction for the olive fruit flies (Table 4). The results show that there was a significant difference between the YSTs with RS, filled with 2-phenethyl acetate, and the YSTs filled with 2-octanone and isoamyl alcohol (*p* = 0.00916 and *p* = 0.00547, respectively). Yellow sticky traps containing RS filled with 2-phenethyl acetate were significantly more attractive to olive fruit flies in an olive grove than the other volatiles tested. Similarly, there was no significant difference in the attraction for the olive fruit flies between YSTs with RS filled with 2-octanone and isoamyl alcohol (*p* = 0.39374) (Table 4).

In contrast to the above results, the YSTs containing PPV loaded with 2-phenethyl acetate were twice as attractive to olive fruit flies as YSTs containing PPV loaded with 2-octanone or isoamyl alcohol (Figure 5). These results are consistent with those of Vitanović et al. [15], in which the authors indicated that isoamyl alcohol added to PPV and attached to YST was twice as attractive to *B. oleae* in an olive grove as YST with PPV containing 2-phenethyl acetate. Davis et al. [26] also showed that isoamyl alcohol was responsible for trapping a large number of other dipterans.

The results also show that YSTs with RS, loaded with 2-phenethyl acetate, were four times more attractive to *B. oleae* than YSTs with PPV containing the same volatile, whereas the opposite was true for the other two OFF-associated yeast volatiles. The number of flies caught on the YSTs with PPV containing 2-phenethyl acetate stopped increasing in the second week, while the number of olive fruit flies caught on the YSTs with PPV containing the other two volatiles tested continued to increase slightly until the end of the study. As with the study of YSTs containing RS as dispensers, there were no differences between the catches of male and female flies on any of the YSTs tested with PPVs during the study (Figure 6).

Three Mann–Whitney U tests were also performed, one for each pair of YSTs with PPVs containing two different OFF-associated yeast volatiles, to determine if there were significant differences in the attraction of the olive fruit flies (Table 4). The results show that there was no significant difference in the attraction for *B. oleae* among the YSTs with PPV containing all tested OFF-associated yeast volatiles (Table 4).

According to the results presented in Table 5, 2-phenethyl acetate proved to be significantly more attractive to olive fruit flies when loaded in RS than when filled in PPV and attached to YST (*p* = 0.0073). The results obtained may be due to the chemical nature of the compound, as well as to a more uniform release from RS than from PPV, since the entire surface is exposed, and the compound is absorbed into the dispenser itself. However, the results of our study also showed that there was no significant difference in attraction of olive fruit flies among the tested dispensers when loaded with the other two OFF-associated yeast volatile compounds (*p* = 0.1964 and *p* = 0.0603 for isoamyl alcohol and 2-octanone, respectively) (Table 5). In addition to examining the general attraction of olive fruit flies, the difference between the total number of females and males during the four weeks of the experiment was checked (Figure 6). It is well known that conventional attractants mainly attract only one sex: males [27]; thus, we can emphasize that this type of attractant is suitable for attracting both sexes.

Previous studies have shown that 2-phenethyl acetate attracts the olive fruit fly both in the olive grove and in the laboratory [2,3]. Since the differences in the attraction of *B. oleae* in the olive grove have not yet been investigated with different types of dispensers attached to YSTs from which the abovementioned volatile substance is released, our study was the first of its kind. The results of our study show the advantage of using RS on YST loaded with 2-phenethyl acetate as the most attractive bait for olive fruit fly among all the baits studied. For all of these reasons, the results of our study can contribute to a better understanding of all factors used to monitor and/or control *B. oleae*, including the most attractive volatile compound and the most effective dispenser.

### 2.3. The Influence of Climatic Parameters on the Attraction of the Olive Fruit Fly

Various climatic parameters influence the attraction of olive fruit fly by YST-containing volatile compounds as attractants in two ways: first, by the evaporation of volatiles from dispensers and, second, by the population and development of *B. oleae*. Of all climatic parameters, temperature and humidity are the most important for the development of most insect species, including the olive fruit fly [17,28,29]. On the other hand, the evaporation of volatiles from dispensers depends not only on climatic conditions, such as temperature, humidity, air pressure, and wind speed, but also on the chemical nature of the compound and the type of dispenser from which it evaporates. 

Figure 7 shows the climatic conditions during the field bioassay and the number of olive fruit flies caught on YSTs containing RS filled with 2-phenethyl acetate, the volatile compound that was the most attractive to *B. oleae* of all the volatiles tested in the olive grove.

As mentioned above, 2-phenethyl acetate was the most stable of all the volatile compounds studied, evaporated the slowest, and was significantly negatively correlated with temperature under field conditions. In addition to temperature, the concentration of the above volatile compound was also negatively correlated with precipitation, cloud cover, and wind, while it was positively correlated with humidity and air pressure (Table 3).

The highest attraction of olive fruit fly to the YSTs containing RS, filled with 2-phenethyl acetate, was observed in the second and third weeks, while the lowest attraction was recorded in the first week of the study. The intensity of the attraction of *B. oleae* to the mentioned traps can be related to the decrease in the 2-phenethyl acetate concentration. These results can be related to the results of Vitanović et al. [15], who showed that the lowest concentrations of 2-phenethyl acetate elicited a better response from olive fruit fly in the laboratory bioassay. Even though temperatures in the first week of the study (up to 22.2 °C) were more optimal for olive fruit fly development [30,31], the fly capture was lower than in the other weeks of the study (15.3–19.3 °C), which could be due to the negative correlation between 2-phenethyl acetate and temperature (Table 3). This implies that temperature has a major influence on the evaporation of certain volatiles and the attraction of *B. oleae*.

In addition to temperature, olive fruit fly development is highly dependent on humidity [32,33], with an optimum of 55–75% [32], which may also affect the effectiveness of olive fruit fly monitoring and/or control methods [34,35]. This fact is also confirmed by the results of our study, because the indicated optimum humidity was determined exactly on the days when *B. oleae* was intensively trapped (Figure 7). In addition, the results of our study showed that 2-phenethyl acetate is positively correlated with humidity (Table 3), which means that its evaporation is associated with a higher percentage of humidity. The results of our study confirmed this, as the YSTs with RS, loaded with 2-phenethyl acetate, exerted a better attraction of olive fruit flies when the highest humidity was measured. Moreover, on the days when the tested traps exerted a high attraction of olive fruit flies, a moderate wind prevailed (Figure 7), which may have influenced the release of the tested volatile compounds [20]. It can be assumed that the wind speed was negligible during the study and had no effect on fly capture, since the evaporation of 2-phenethyl alcohol is negatively correlated with wind (Table 3) and wind has a negative effect on *B. oleae* flight. Finally, the results of our study show the inevitable influence of the high variability of field conditions on the behavior of the chemical in the dispenser and, thus, on its attraction of the olive fruit fly. Under such uncontrolled conditions, it is difficult to infer the trigger of the biological activity of the volatiles, especially since their synergistic or antagonistic effect is certainly present.

## 3. Materials and Methods

### 3.1. Synthetic Volatile Compounds

All chemical standards used for this study (Table 1) were obtained from commercial sources and had a purity of ≥99%. Isoamyl alcohol was obtained from Kemika d.d. (Zagreb, Croatia), 2-phenethyl acetate from Sigma-Aldrich (St. Louis, MO, USA), 2-octanone from Alpha Aesar (Kandel, Germany), n-hexane from VWR International bvba (Leuven, Belgium), and acetone from Kemika d.d. (Zagreb, Croatia).

### 3.2. Slow-Release Dispensers

The release rate of all volatile compounds tested was determined using two types of slow-release dispensers: 4 mL polypropylene vials (PPVs) (Cryotubes, BRAND GMBH + CO KG, Wertheim, Germany) with a 3 mm diameter hole in the lids and rubber septa (RS) (2.4 mm × 5.33 mm, Sigma Aldrich, St. Louis, MO, USA). A total of 0.1 g of cotton and 1 mL of each synthetic volatile compound tested was added to the bottom of a 4 mL PPV [15,24]. The purified RS were loaded with 50 µL of each synthetic volatile compound tested. To absorb the volatile compounds, the loaded RS were placed in a filter fume hood (GS1500; Gruppo Strola, Torino, Italy) for 24 h to absorb the volatile compounds before performing the experiments. Prior to volatile loading, all septa were prepared and cleaned by Soxhlet extraction with n-hexane for 24 h, followed by methylene chloride for 24 h and air-drying in a fume hood for an additional 24 h [20,35]. The cleaned and sealed septa were stored at 4 °C and then subjected to HS-GC/FID analysis to prove the cleanliness of the dispenser and to rule out the presence of interfering substances that could affect further release rate analyses, as well as to rule out interference with OFF field attraction.

### 3.3. Experiment Design

The study was conducted in October 2021 and consisted of two separate experiments, as shown in the flowchart (Figure 1). In experiment 1, the release ratio of the OFF-associated yeast was studied by measuring the mass change in the RS and PPV and by measuring the residual volatiles in the RS over 4 weeks under controlled and field conditions. In experiment 2, conducted in an olive orchard, the attraction of olive fruit flies to the volatile compounds of the OFF-associated yeasts was studied.

#### Monitoring of the Release Rate of the Tested Volatile Compounds: Experiment 1

The study of the release ratio of the OFF-associated yeast volatile compounds (Experiment 1) consisted of three independent measurements performed under different environmental conditions (i.e., room conditions, environmental chamber conditions, and field conditions) to estimate the release ratio of three OFF-associated yeast volatile compounds (isoamyl alcohol, 2-octanone, and 2-phenethyl acetate) and a non-yeast volatile compound (n-hexane), which was used as a positive control, from two types of dispensers, RS and PPV. To monitor the release ratio of the OFF-associated yeast compounds, the tested RS and PPV were weighed using an analytical balance (Mettler Toledo, OH, USA) before the addition of the volatile compound, 24 h after the addition of the volatile compound, and daily at the same time and under the same conditions to measure the weight loss over the 4 week experiment. To test the presence of the tested volatiles using HS-GC/FID, the rubber septa were placed in triplicate on each measurement day after washing and loading in all environmental conditions tested. The release rate of each tested volatile compound placed in dispensers (50 µL in RS and 1 mL in PPV) was examined individually.

The climatic parameters of the environmental conditions measured in the room environment, in the environmental chamber, and in the field are shown in Table 6. The parameters of the environmental conditions in the room were recorded with wireless sensors (Agara temperature and humidity sensor). The environmental chamber (Kambič, Semič, Slovenia) with the central microprocessor control DPC-420 was used to simulate the conditions in the field and to eliminate external influences (mainly temperature fluctuations, wind, solar radiation, precipitation, etc.). A temperature range was set to simulate the average daytime (23 ± 1 °C) and nighttime (17 ± 1 °C) temperatures throughout the experiment. At the same time, the influence of the actual climatic parameters on the release ratio of all studied volatile compounds in the olive grove was tested using the weather data from the nearest weather station [29] (see Table 6).

The measurement of the presence of the tested chemicals was performed using two different methods: measuring the changes in the mass and concentration of the individually tested chemicals on the same days (1st, 2nd, 3rd, 4th, 5th, 6th, 7th, 9th, 10th, 11th, 12th, 13th, 14th, 17th, 21st, and 28th days) in triplicate over a 4 week experiment under controlled and field conditions. The mass measurement was performed using an analytical balance (Mettler Toledo, Ohio, SAD) for the compounds loaded in RS and PPV, while the measurement of the residual amount of volatile compounds loaded in RS was performed using HS-GC/FID (Shimadzu, Kyoto, Japan).

### 3.4. Field Bioassay: Experiment 2

The second separate but simultaneous experiment (Experiment 2, Figure 2) was conducted in the olive grove of the Institute for Adriatic Crops (IAC) at the Duilovo site, Split, Croatia (geographic coordinates: 43°30′19.4′′ N 16°29′56.1′′ E), elevation 73 m a.s.l. The olive grove was regularly maintained, except for plant protection measures. During the 4 week experiment, climate [36] data (temperature, humidity, wind speed, precipitation, air pressure, and cloud cover) were collected from the Split airport site [36]. Hourly measurements were aggregated to daily values as follows: air temperature, relative humidity, and air pressure were averaged; wind speed, precipitation, and cloud cover were summed (each parameter separately). Each parameter was then normalized and plotted on a stacked graph. The number of OFFs was divided by the number of days between two consecutive measurements to determine the daily number of OFFs.

#### Olive Fruit Flies Trapping

The olive fruit fly attraction to OFF-associated yeast volatile compounds was tested using traps consisting of double-sided yellow sticky traps (YST) (17 × 24 cm, Bio Plantella, UNICHEM d.o.o.) and two types of slow-release chemical dispensers, 4 mL PPVs and RS, loaded with 3 OFF-associated yeast volatile compounds, and a positive control, n-hexane. The dispensers were prepared for the experiment following the same protocol as described in Section 3.2. The PPVs, loaded with 1 mL, were attached to the YS traps with cable ties, while the RS, loaded with 50 µL of the tested volatile compound, were placed in a PVC-coated fiberglass insect net and attached to the YSTs. The trap experiment included 8 treatments in triplicate (Table 7).

Yellow sticky traps containing PPVs and RS were placed in a random arrangement evenly distributed in the olive orchard. No modifications were made to the slow-release chemical dispensers containing synthetic OFF-associated yeast volatiles. The traps were placed at a height of 1.5 to 2 m above the ground and in a southwesterly direction between every other row in the olive orchard at a distance of ≈15 m within the row. To observe whether the installed dispensers attracted OFFs over the 4 week experiment, the OFFs were counted and removed several times (fourteen) at different time intervals.

### 3.5. HS-GC/FID Analysis

The measurement of the amount present of all tested volatile compounds (3 OFF-associated yeast volatile compounds and 1 non-yeast volatile compound) in RS was performed using a gas chromatograph (Nexis GC-2030, Shimadzu, Japan) coupled with a headspace and flame ionization detector (HS-GC/FID). The amount of each tested volatile compound (50 µL in RS) was determined by an in-house method using a SH-RTX-WAX capillary column (30 m × 0.25 mm × 0.25 µm) and quantified by comparing their retention time with the retention time of a standard under the same analysis conditions. The measurements were performed in triplicate. Ultra-high purity helium (99.999% purity) with a constant flow of 1 mL/min was used as the carrier gas. The oven temperature was set at 40 °C, held for 1 min, increased to 225 °C at a rate of 30 °C/min, and then maintained for an additional minute. The analysis time was 8.17 min. Headspace conditions were set as follows: thermostating at 80 °C for 5 min with a rotation speed of 250 rpm and a needle transfer temperature of 105 °C. The injection temperature was set at 200 °C. The analysis results are expressed as the peak area and as the mass concentration, calculated using calibration curves, using 6 calibration levels for each tested volatile compound, in a linear range from 0.033 to 10.32 ppm, where linearity, the coefficient of determination (R^2^), slope, and intercept were assessed (Appendix A). Acetone was used as an internal standard at a concentration of 0.01 ppm. All analysis was performed in triplicate.

### 3.6. Data Analysis

#### 3.6.1. Pearson Correlation Coefficient

To determine whether there was a correlation between the change in the mass of the OFF-associated yeast volatile compounds in RS and PPV under the environmental conditions tested and a correlation between the concentration of the OFF-associated yeast volatile compounds in the rubber septa and the climatic parameters during the study, a Pearson correlation coefficient analysis was performed. The correlation was considered significant at a value of *p* ≤ 0.05. The analysis was performed using SPSS software, version 25.0 (IBM Corporation, New York, NY, USA).

#### 3.6.2. Mann–Whitney U Test

The statistical analysis was performed using the Python programming language [37]. The experimental data were fitted using the nonlinear least squares method from the scipy package. The mean values of the number of flies for each volatile compound tested were compared with the Mann–Whitney U nonparametric test at a significance level of *p* < 0.05 using the Stats models package to test whether the number of OFFs attracted to different volatile compounds represented a different population with different mean values. Since the samples were not normally distributed, a nonparametric alternative to the parametric two-sample t-test was chosen.

## 4. Conclusions

To enable the manipulation of OFF, the knowledge of the physicochemical properties of semiochemicals and the dispensers used to release them must be expanded to better mimic natural environmental conditions. In our study, we confirmed the presence of all tested OFF-associated yeast volatile compounds by both methods: mass weighing and gas chromatography. Since measuring the mass of the septa with the small amounts of volatiles used in this experiment was not accurate enough to confirm their presence, a more precise analytical technique, HS-GC/FID, was used. This technique was able to confirm the presence of all tested volatiles in the dispensers, which may allow for the longer use of the dispensers, reducing costs and workforce requirements, as dispensers in olive groves need to be replaced less frequently. The attraction of *B. oleae* to the tested OFF-associated yeast volatile compounds attached to YS traps with both types of dispensers was also demonstrated, with 2-phenethyl acetate loaded in RS showing a statistically significant difference in attracting *B. oleae*. The results of this study suggest that further experiments with OFF-associated yeast volatile compounds are needed when used in passive controlled-release dispensers to attract olive fruit flies. Testing different combinations of volatiles in different dispensers, both under natural conditions and at different concentrations, requires further attention.

## Figures and Tables

**Figure 1 molecules-28-02431-f001:**
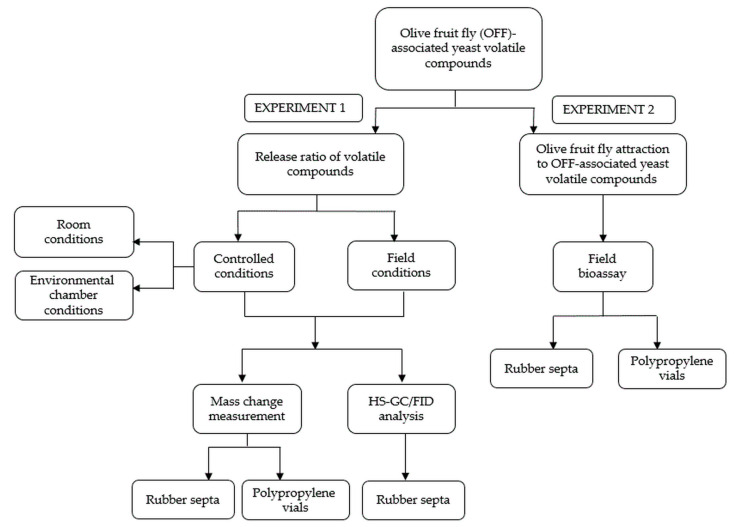
Flowchart of the study.

**Figure 2 molecules-28-02431-f002:**
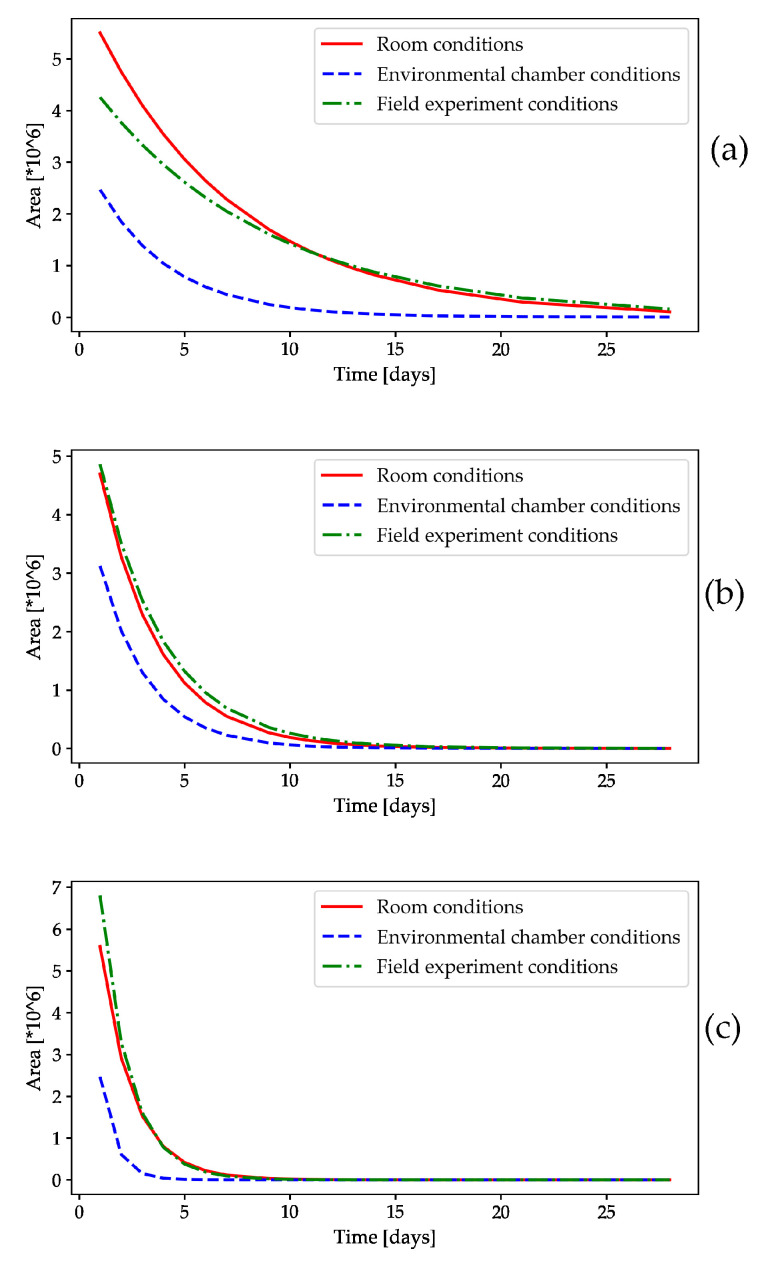
Daily release of the individual OFF-associated yeast volatile compounds: (**a**) isoamyl alcohol, (**b**) 2-octanone, and (**c**) 2-phenethyl acetate analyzed under controlled and field conditions using HS-GC/FID.

**Figure 3 molecules-28-02431-f003:**
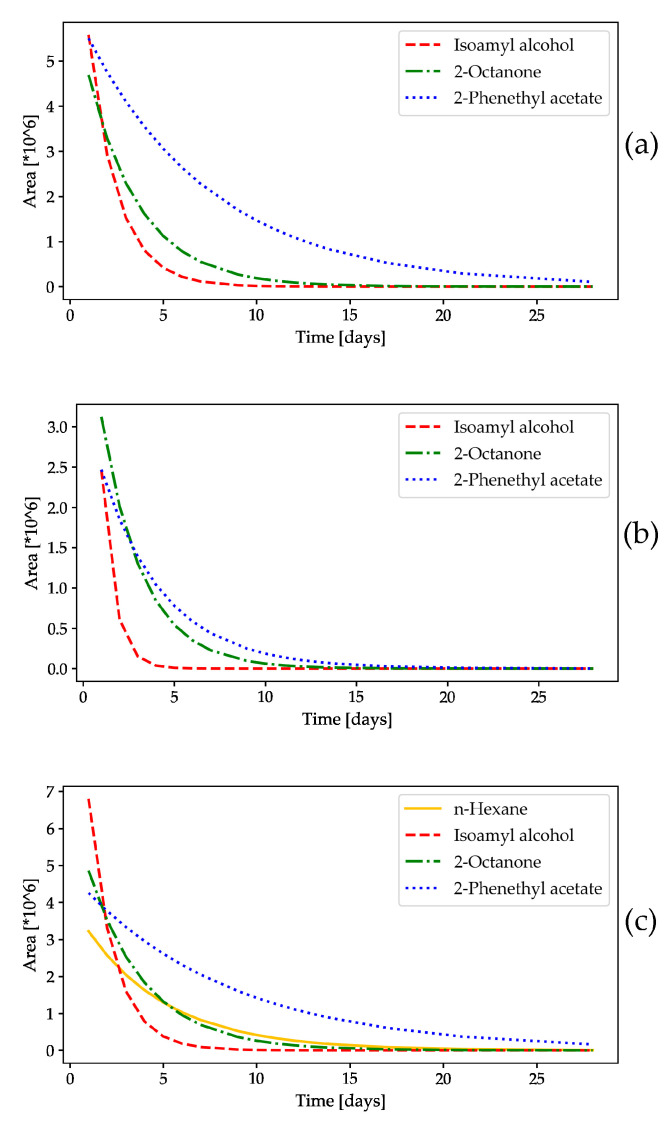
Daily release of volatile compounds under the following conditions: (**a**) room conditions (3 OFF-associated yeast volatile compounds); (**b**) environmental chamber conditions (3 OFF-associated yeast volatile compounds); (**c**) field conditions (3 OFF-associated yeast volatile compounds and n-hexane), analyzed by HS-GC/FID.

**Figure 4 molecules-28-02431-f004:**
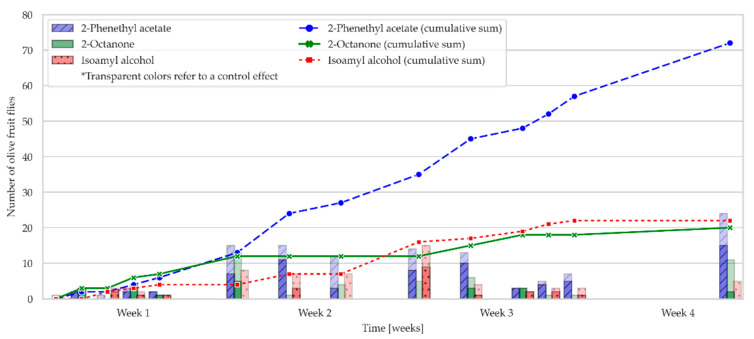
The figure shows the attraction of OFFs towards the tested olive fruit fly-associated yeast compounds (isoamyl alcohol, 2-octanone, and 2-phenethyl acetate) in rubber septa over a 4 week experiment. The total number of OFFs is represented by the total height of the bars. The transparent parts of the bars represent the control effect (number of OFFs attracted to n-hexane). The fitted values are represented as colored bars. The cumulative sums of the fitted values are shown as lines.

**Figure 5 molecules-28-02431-f005:**
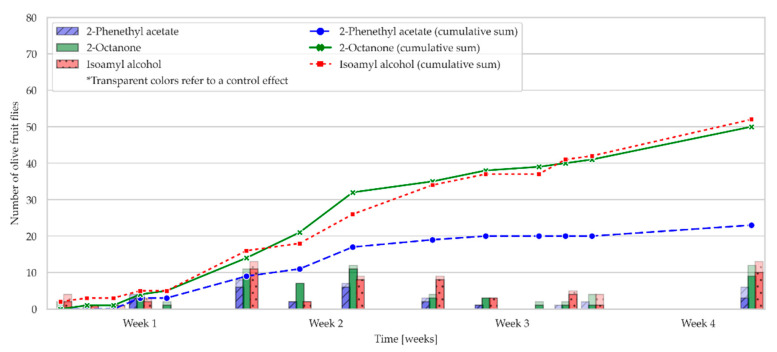
The figure shows the attraction of OFFs to the tested olive fruit fly-associated yeast compounds (isoamyl alcohol, 2-octanone, and 2-phenethyl acetate) in polypropylene vials over a 4 week experiment. The total number of OFFs is represented by the total height of the bars. The transparent parts of the bars represent the control effect (number of OFFs attracted to n-hexane). The fitted values are represented as colored bars. The cumulative sums of the fitted values are shown as lines.

**Figure 6 molecules-28-02431-f006:**
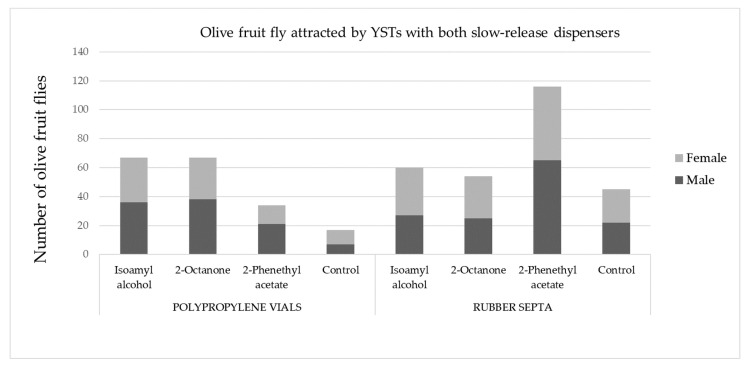
Total number of male and female olive fruit flies attracted to the tested volatile compounds (3 OFF-associated yeast compounds and control) attached to YST in PPV and RS.

**Figure 7 molecules-28-02431-f007:**
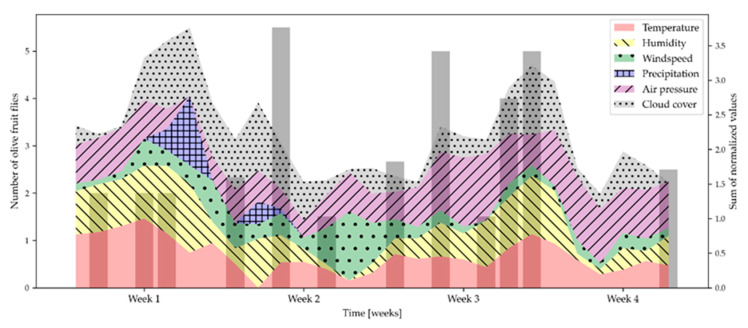
The attraction of OFFs to 2-phenethyl acetate in rubber septa under various climatic conditions (temperature, humidity, wind speed, precipitation, air pressure, and cloud cover), measured hourly, aggregated to daily values, and summed to normalized values.

**Table 1 molecules-28-02431-t001:** Volatile compounds tested in the study.

Compound	Chemical Structure	Molecular Formula	Density(g/cm^3^)	Molecular Weight (g/mol)	Classification
Isoamyl alcohol	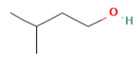	C_5_H_12_O	0.810	88.15	Primary ALCOHOL
2-Octanone	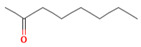	C_8_H_16_O	0.820	128.21	Methyl KETONE
2-Phenethyl acetate	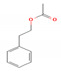	C_10_H_12_O_2_	1.032	164.20	Acetate ESTER
n-Hexane	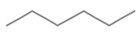	C_6_H_14_	0.655	86.18	Straight-chain ALKANE

Table data were sourced from PubChem for each compound individually [20].

**Table 2 molecules-28-02431-t002:** Correlation between the measured mass changes of all tested volatiles in rubber septa and polypropylene vials during the experiment.

Tested Volatile	Room Conditions	Environmental Chamber Conditions	Field Conditions
Isoamyl alcohol	*r* = 0.2605	*r* = 0.4418	*r* = 0.2739
	*p* = 0.2814	*p* = 0.0582	*p* = 0.2565
2-Octanone	*r* = 0.3446	***r* = 0.5139**	*r* = 0.3149
	*p* = 0.1485	***p* = 0.0244**	*p* = 0.1891
2-Phenethyl acetate	***r* = 0.557**	***r* = 0.647**	***r* = 0.5774**
	***p* = 0.0133**	***p* = 0.0026**	***p* = 0.0096**
n-Hexane	***r* = 0.6923**	***r* = 0.7141**	***r* = 0.7179**
	***p* = 0.0010**	***p* = 0.0059**	***p* = 0.0005**

*r* = Pearson’s correlation coefficient; correlation was considered significant at a level of *p* ≤ 0.05; absolute linear correlation coefficients ≥ |0.50| are marked in bold.

**Table 3 molecules-28-02431-t003:** The correlation between the climatic parameters and the concentration of the tested volatile compounds in rubber septa under all tested environmental conditions during the experiment.

		Climatic Parameters					
Environmental Conditions	Volatile Compound	Temperature (°C)	Relative Humidity (%)	Air Pressure (hPa)	Precipitation (mm)	Cloud Cover (okta)	Wind (Bf)
Room conditions	Isoamyl alcohol	*r* = −0.3727	*r* = 0.2845	*r* = 0.346	Not analyzed		
		*p* = 0.1560	*p* = 0.2855	*p* = 0.1893			
	2-Octanone	*r* = −0.399	*r* = 0.3299	*r* = 0.3295	Not analyzed		
		*p* = 0.1258	*p* = 0.2121	*p* = 0.2127			
	2-Phenethyl acetate	***r*** **= −0.8269**	*r* = 0.3378	*r* = 0.2646	Not analyzed		
		***p*** **= 0.0001**	*p* = 0.2007	*p* = 0.3225			
Environmental chamber conditions	Isoamyl alcohol	*r* = −0.2906	*r* = 0.1582	*r* = 0.2887	Not analyzed		
		*p* = 0.2759	*p* = 0.5584	*p* = 0.2872			
	2-Octanone	*r* = −0.3201	*r* = 0.3678	*r* = 0.3738	Not analyzed		
		*p* = 0.2269	*p* = 0.1610	*p* = 0.1538			
	2-Phenethyl acetate	***r*** **= −0.6483**	*r* = 0.384	*r* = 0.3606	Not analyzed		
		***p*** **= 0.0066**	*p* = 0.1420	*p* = 0.1700			
Field conditions	Isoamyl alcohol	*r* = −0.4246	*r* = 0.2665	*r* = 0.2773	*r* = −0.1217	*r* = −0.2321	*r* = −0.2569
		*p* = 0.1152	*p* = 0.3184	*p* = 0.3170	*p* = 0.6553	*p* = 0.3873	*p* = 0.3386
	2-Octanone	*r* = −0.4253	*r* = 0.3713	*r* = 0.2638	*r* = −0.0604	*r* = −0.2403	*r* = −0.2679
		*p* = 0.1108	*p* = 0.1568	*p* = 0.3235	*p* = 0.8253	*p* = 0.3706	*p* = 0.3175
	2-Phenethyl acetate	***r*** **= −0.6632**	*r* = 0.2679	*r* = 0.2315	*r* = −0.0414	*r* = −0.2384	*r* = −0.0927
		***p*** **= 0.0051**	*p* = 0.3158	*p* = 0.3883	*p* = 0.8802	*p* = 0.3747	*p* = 0.7347
	n-Hexane	*r* = −0.4989	*r* = 0.3322	*r* = 0.2077	*r* = 0.0201	*r* = −0.2146	*r* = −0.1801
		***p*** **= 0.0496**	*p* = 0.2087	*p* = 0.4402	*p* = 0.9401	*p* = 0.4261	*p* = 0.5047

*r* = Pearson’s correlation coefficient; correlation was considered significant at the level of *p* ≤ 0.05; absolute linear correlation coefficients ≥ |0.50| are marked in bold.

**Table 4 molecules-28-02431-t004:** Comparison of the attractiveness of olive fruit fly-associated yeast volatiles in slow-release dispensers attached to yellow sticky traps to *Bactrocera oleae*.

	Compared OFF-Associated Yeast Volatiles	U Statistic	*p*-Value
Polypropylene vials	2-Phenethyl acetate, 2-octanone	44.0	0.05213
	2-Phenethyl acetate, isoamyl alcohol	50.5	0.10561
	2-Octanone, isoamyl alcohol	68.5	0.43005
Rubber septa	2-Phenethyl acetate, 2-octanone	31.5	0.00916
	2-Phenethyl acetate, isoamyl alcohol	28.0	0.00547
	2-Octanone, isoamyl alcohol	67.0	0.39374

Correlation was considered significant at the level of *p* ≤ 0.05.

**Table 5 molecules-28-02431-t005:** Comparison of the attraction of the yeast volatile compounds associated with olive fruit flies to *Bactrocera oleae* in rubber septa and polypropylene vials attached to yellow sticky traps.

OFF-Associated Yeast Volatile Compound	U Statistic	*p*-Value
2-Phenethyl acetate	30.0	0.00736
Isoamyl alcohol	57.0	0.19644
2-Octanone	45.5	0.06039

Correlation was considered significant at the level of *p* ≤ 0.05.

**Table 6 molecules-28-02431-t006:** Climatic parameters of the environmental conditions during the study.

		Temperature (°C)			Air Pressure (hPa)			Humidity (%)		
Conditions	Week	Min.	Max.	Mean	Min.	Max.	Mean	Min.	Max.	Mean
Room conditions	1st	21.1	23.6	22.76	997	1012	1006.63	48.7	63	56.33
	2nd	17.2	21.2	18.95	1002	1012	1006.17	39.1	52.9	45.89
	3rd	17.1	20.8	18.45	1003	1020	1012.44	43.1	67.6	51.31
	4th	17.4	20	18.54	1008	1019	1013.19	46.2	61.8	51.05
Environmental chamber conditions	1st–4th	17	23	20	1013.25	1013.25	1013.25	60	70	65
Field conditions	1st	15.2	24.3	19.21	1007.91	1021.65	1016.55	42	95	70.96
	2nd	10.6	19.3	14.2	101.71	1020.58	1015.47	29	89	47.69
	3rd	10.3	20.8	16.54	1012.05	1028.31	1020.83	37	89	61.12
	4th	9.4	21.2	15.28	1017.25	1026.85	1021.59	27	90	52.25

**Table 7 molecules-28-02431-t007:** Treatments used in the trapping experiment.

Treatment	Trap (Dispenser Type)	Attractants (Volatile Compounds)	Retention System
A	YST + PPV	Isoamyl Alcohol	Sticky
B	YST + PPV	2-Octanone	Sticky
C	YST + PPV	2-Phenethyl acetate	Sticky
D	YST + PPV	n-Hexane	Sticky
E	YST + RS	Isoamyl Alcohol	Sticky
F	YST + RS	2-Octanone	Sticky
G	YST + RS	2-Phenethyl acetate	Sticky
H	YST + RS	n-Hexane	Sticky

## Data Availability

All data are included within the article and in the Appendix A.

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
