# Peer review of "A Lifetime of a Dispenser-Release Rates of Olive Fruit Fly-Associated Yeast Volatile Compounds and Their Influence on Olive Fruit Fly (Bactrocera oleae Rossi) Attraction"

_molecules, 2023, doi:10.3390/molecules28062431_

Round 1
Reviewer 1 Report
This manuscript details a study of the release rate, and consequently the longevity, of a dispenser of yeast volatiles (lure) which are associated with the olive fruit fly (OFF, Bactrocera oleae Rossi), an insect of high commercial significance to olive growers. This also sheds light on the performance of lures (based on rubber septa, RS, and polypropylene vials, PPV) in the field, and under controlled conditions, and the release rate of OFF attractants is actually quantified. This offers an important perspective on what dispensers do (how much attractant they actually emit) in the field, as a result of abiotic factors such as the physicochemical properties of the semiochemical and atmospheric conditions.
Lures were weighed on an analytical balance to determine mass loss and consequently rate of emission; RS were also analyzed by headspace solid-phase microextracton gas chromatography / flame ionization detection (HS-SPME-GC/FID) because weighing could not give an accurate measure of the very small mass losses from RS. The authors note that “…the release rate of OFF-associated yeast volatile compounds has not been studied…” (lines 87-88); also, they authors report that “…RS, loaded with 2-phenethyl acetate, were the most attractive to OFF of all traps tested.” (lines 268-269).
The analytical determination of the release rate from RS using HS-SPME-GC/FID is thorough, well-described and scientifically sound.
I detected no major methodological errors in the manuscript; the experimental design and statistical analysis seem sound.
I have several editorial comments for the authors’ attention, see below:
-line 25, omit “tested”.
-lines 28-29, I would suggest rephrasing to “…in contrast to the attraction of olive fruit flies to polypropylene vials containing these compounds, but without significant difference.”
-lines 45-46, maybe rephrase this sentence to “as the release of a single compound from an organism can elicit a behavioral or physiological response.” The meaning of this sentence in the text is a little unclear.
-line 48, rephrase to “…a trend towards…” not “…a trend of…”
-line 49, maybe “(because of resistance towards chemical measures)” instead of just “(resistance)”
-line 72, “releasing device” not “release device”
-lines 76-77, “Dispensers for the slow release of semiochemicals…” not “Dispensers for slow release semiochemicals…”
-line 117, Table 1: the heading and body of this table were separate in my reviewer’s copy.
-line 119, omit the comma after “both”.
-lines 125-126, I would suggest rephrasing this sentence to “…use of the dispenser in the field for attraction of OFFs. As can be seen from the results…” ie. make it two sentences instead of one.
-line 147-148; n-hexane has the lowest molecular weight of all four of the compounds tested, but it is also very nonpolar, and has no polar functional groups. This would also increase volatility. The authors could make a note of this at this point in the text.
-line 153, “release rate” not “release ratio”? I would think the authors meant the former, not the latter.
-lines 154-158, there is no wind in the environmental chamber (I think) or in the lab; this would influence evaporation rate of the semiochemicals. The authors could note this at this point in the text. However, the authors do note wind as a climatic parameter in Table 3.
-lines 173-174, “…a higher concentration and volume of volatiles does not necessarily attract more OFF.” Instead of “…a higher concentration and volume of volatiles does not mean that higher efficiency is better.” I make this suggestion as an alternate rephrasing of this sentence. If this is not the actual meaning of the sentence on lines 173-174, the authors should clarify this.
-equation (1); all of the terms in the equation could be defined, not just b.
-line 214, omit “tested”.
-line 215, insert a comma between 2 and 3.
-lines 223-224, it is unclear to me what the percentages in the parentheses refer to. The authors quote supplementary figure S1, but a line or two in the text to clarify this would be helpful.
-likewise line 232.
-line 261, I would suggest “accuracy and precision” instead of “correctness”
-line 262, “yeast” not “yeasty” and “in an olive grove” not “in olive grove”
-line 293, “in an olive grove” not “in olive grove”
-line 294, I would suggest rephrasing this sentence to “…between YSTs with RS filled with 2-octanone and isoamyl alcohol.”
-Figures 2 and 3, the text inside the figures showing what the different colored lines are is too small to read in my reviewer’s copy.
-line 427 “…of a 4 mL PPV.” (insert “a”)
-line 436 “analyses” not “analyzes”
-Tables 7 and 8, maybe insert a couple of line spaces between the tables and the body of the text that comes after.
-line 511, I would suggest rephrasing “…the present amount…” to “…the amount present…”
-line 520, probably should be 30°C/min., not 3030°/min.
-line 531, omit “if”.
-line 641, reference 24, put a period at the end, not a semicolon.
Author Response
We would like to thank the reviewer for their affirmative and constructive comments. We hope that we have addressed all your concerns. The changes have been marked in yellow.
- line 25, omit “tested”.
Response: tested is omitted
- lines 28-29, I would suggest rephrasing to “…in contrast to the attraction of olive fruit flies to polypropylene vials containing these compounds, but without significant difference.”
Response: the sentence has been rephrased; see lines 29 and 30
- lines 45-46, maybe rephrase this sentence to “as the release of a single compound from an organism can elicit a behavioral or physiological response.” The meaning of this sentence in the text is a little unclear.
Response: the sentence has been rephrased; see lines 46 and 47
- line 48, rephrase to “…a trend towards…” not “…a trend of…”
Response: rephrased; see line 49
- line 49, maybe “(because of resistance towards chemical measures)” instead of just “(resistance)”
Response: The sentence is remodeled in the text and marked. It would not be all right to say "according to chemical measures" since semiochemicals are also chemical measures on the one hand, except that they are more environmentally friendly for different organisms, the environment and humans; see lines 50 and 51
- line 72, “releasing device” not “release device”
Response: Corrected; see line 74
- lines 76-77, “Dispensers for the slow release of semiochemicals…” not “Dispensers for slow release semiochemicals…”
Response: corrected; see lines 78 and 79
- line 117, Table 1: the heading and body of this table were separate in my reviewer’s copy.
Response: The table is corrected; see lines 120 and 121
- line 119, omit the comma after “both”.
Response: “both” is omitted
- lines 125-126, I would suggest rephrasing this sentence to “…use of the dispenser in the field for attraction of OFFs. As can be seen from the results…” ie. make it two sentences instead of one.
Response: The sentence is corrected; see lines 128 and 129
- line 147-148; n-hexane has the lowest molecular weight of all four of the compounds tested, but it is also very nonpolar, and has no polar functional groups. This would also increase volatility. The authors could make a note of this at this point in the text.
Response: We added a note about this according to your comment; see line 151 and 152
- line 153, “release rate” not “release ratio”? I would think the authors meant the former, not the latter.
Response: Corrected to release rate; see line 155
- lines 154-158, there is no wind in the environmental chamber (I think) or in the lab; this would influence evaporation rate of the semiochemicals. The authors could note this at this point in the text. However, the authors do note wind as a climatic parameter in Table 3.
Response: The sentence is corrected and marked. Also, controlled conditions have no wind, precipitation and similar climate parameters. These parameters are measured only for field experiment, and expressed in Table 3.; see lines 162 and 163
- lines 173-174, “…a higher concentration and volume of volatiles does not necessarily attract more OFF.” Instead of “…a higher concentration and volume of volatiles does not mean that higher efficiency is better.” I make this suggestion as an alternate rephrasing of this sentence. If this is not the actual meaning of the sentence on lines 173-174, the authors should clarify this.
Response: The sentence is corrected according to comment; see line 180
- equation (1); all of the terms in the equation could be defined, not just b.
Response: The explanation of the terms in the equation are added; see lines 200 and 201
- line 214, omit “tested”.
Response: tested is omitted
- line 215, insert a comma between 2 and 3.
Response: a comma has been added; see line 222
- lines 223-224, it is unclear to me what the percentages in the parentheses refer to. The authors quote supplementary figure S1, but a line or two in the text to clarify this would be helpful.
Response: Thank you for your suggestion. The sentence is remodeled to be more clear; that is, the percentages relate to the part that is left of the initial volume; see lines 228-234
- likewise line 232.
Response: the sentence was corrected; see lines 241-243
- line 261, I would suggest “accuracy and precision” instead of “correctness”
Response: correctness is replaced with accuracy and precision; see line 272
- line 262, “yeast” not “yeasty” and “in an olive grove” not “in olive grove”
Response: corrections have been made; see line 274
- line 293, “in an olive grove” not “in olive grove”
Response: corrections have been made; see line 305
- line 294, I would suggest rephrasing this sentence to “…between YSTs with RS filled with 2-octanone and isoamyl alcohol.”
Response: the sentence is rephrased; see line 306
- Figures 2 and 3, the text inside the figures showing what the different colored lines are is too small to read in my reviewer’s copy.
Response: Figures are enlarged and improved; Figure 2 (line 212), Figure 3 (line 216) and Figure 6 (line 359)
- line 427 “…of a 4 mL PPV.” (insert “a”)
Response: “a” is added; see line 435
- line 436 “analyses” not “analyzes”
Response: the correction has been made; see line 447
- Tables 7 and 8, maybe insert a couple of line spaces between the tables and the body of the text that comes after.
Response: line spaces are added; lines 483 and 515
- line 511, I would suggest rephrasing “…the present amount…” to “…the amount present…”
Response: the correction has been made; see line 524
- line 520, probably should be 30°C/min., not 3030°/min.
Response: the correction has been made; see line 533
- line 531, omit “if”.
Response: the correction has been made
- line 641, reference 24, put a period at the end, not a semicolon.
Response: the correction has been made; see line 653

Reviewer 2 Report
Title
It would be better to choose a more effective and short title.
Abstract
Using a sentence at the beginning abstract which includes the research problem and the importance of this study.
Keywords
Olive, olive fly, volatile compounds, semiochemicals, slow-release dispensers, and yeast terms are enough for further citations.
Introduction
Line 50: instead of ‘olive fruit fly (Bactrocera oleae, Rossi) (Diptera: Tephritidae) (OFF)’ please prefer ‘olive fruit fly (OFF) namely Bactrocera oleae, Rossi’.
Materials and Methods
Please use regular ‘academic writing format’.
1. Introduction,
2. Material and Methods
3. Results
4. Discussion
5. Conclusion
In this manuscript, results are coming before the material & method section.
Statistics
Appropriate.
Results and Discussion
Please reduce the number of Tables.
Please improve the image quality and their presentations.
References
Please check the reference style of the journal and rearrange the typos.
Others:
Please use the abbreviations correctly.
Proofreading is essential for this manuscript.
Author Response
Thank you for the effort in reviewing and improving our manuscript paper. We have considered all the comments and suggestions and the manuscript has been modified accordingly. The changes have been marked in yellow.
- Title
It would be better to choose a more effective and short title.
Response: Thank you very much for your suggestion, but shortening the title would mean excluding the important parts covered by this research.
- Abstract
Using a sentence at the beginning abstract which includes the research problem and the importance of this study.
Response: The sentence is corrected; see line 20
- Keywords
Olive, olive fly, volatile compounds, semiochemicals, slow-release dispensers, and yeast terms are enough for further citations.
Response: Thank you for the suggestion, keywords have been changed; see lines 33-34
- Introduction
Line 50: instead of ‘olive fruit fly (Bactrocera oleae, Rossi) (Diptera: Tephritidae) (OFF)’ please prefer ‘olive fruit fly (OFF) namely Bactrocera oleae, Rossi’.
Response: Corrections have been added in accordance with the instructions; see line 52
- Materialsand Methods
Please use regular ‘academic writing format’.
- Introduction,
- Material and Methods
- Results
- Discussion
- Conclusion
In this manuscript, results are coming before the material & method section.
Response: Thank you for the suggestion, but the Journal's propositions in the "Research Manuscript Section"( https://www.mdpi.com/journal/molecules/instructions#preparation) state that the correct order when sending manuscript would be: Introduction, Results, Discussion, Materials and Methods, Conclusion, etc.; so we prepared our manuscript according to those instructions.
- Statistics
Appropriate.
- Results and Discussion
Please reduce the number of Tables.
Response: Tables 4 and 5 were merged into Table 4; see lines 308-310
Please improve the image quality and their presentations.
Response: Figures were improved; Figure 2 (line 212), Figure 3 (line 216) and Figure 6 (line 359)
- References
Please check the reference style of the journal and rearrange the typos.
Response: References were checked and corrected according to the journal's propositions
- Others:
Please use the abbreviations correctly.
Response: We checked the manuscript and to our best knowledge, all the abbreviations are correct.
Proofreading is essential for this manuscript.
Response: The manuscript has undergone proofreading; corrections have been added as can be seen in the manuscript
